# The Protective Role of the FIFA 11+ Training Program on the Valgus Loading of the Knee in Academy Soccer Players Across a Season

**DOI:** 10.3390/healthcare13010073

**Published:** 2025-01-03

**Authors:** Michele Mercurio, Giovanni Carlisi, Marko Ostojic, Alessandro Imbrogno, Olimpio Galasso, Giorgio Gasparini

**Affiliations:** 1Department of Orthopaedic and Trauma Surgery, Magna Graecia University, R. Dulbecco University Hospital, 88100 Catanzaro, Italy; giovanni.carlisi@studenti.unicz.it (G.C.); alessandro.imbrogno@studenti.unicz.it (A.I.); gasparini@unicz.it (G.G.); 2Research Center on Musculoskeletal Health, MusculoSkeletalHealth@UMG, Magna Graecia University, 88100 Catanzaro, Italy; 3Osteon Orthopedics and Sports Medicine Clinic, 88000 Mostar, Bosnia and Herzegovina; marko.ostojic@ymail.com; 4Department of Medicine, Surgery and Dentistry, University of Salerno, 84081 Baronissi, Italy; ogalasso@unisa.it

**Keywords:** dynamic valgus angle, drop vertical jump test, anterior cruciate ligament injury, landing, pivoting, FIFA 11+ program

## Abstract

**Background**: Improper neuromuscular control with excessive dynamic valgus loading of the knee has been identified as one of the main anterior cruciate ligament injury risk factors. This study aimed to analyze the impact of the FIFA 11+ training program on the valgus loading of the knee in academy soccer players over a competitive season. **Methods**: A prospective study was conducted on 85 players. The drop vertical jump test was carried out before the match and at the end of the same match at the beginning and at the end of the season over a period of 11 months. **Results**: An increase of the varus angle on the right limb was noted between the start and the end of the season at the beginning of the match (−4.7 ± 8.9 versus −6.9 ± 6, *p* = 0.003) and between the start and the end of the match in values measured at the beginning of the season (−4.7 ± 8.9 versus −7.7 ± 9, *p* < 0.001). An increase of the flexion angle of both limbs was noted between the start and the end of the season in values measured at the start of the match (left limb 76.8 ± 32.8 versus 98.6 ± 17.2, *p* < 0.001; right limb 76.4 ± 32.8 versus 96.1 ± 16.1, *p* < 0.001) and between the start and the end of the season in values measured at the end of the match (left limb 92.8 ± 19.1 versus 98.5 ± 16.3, *p* = 0.002; right limb 92.6 ± 19.2 versus 96.7 ± 14.5, *p* = 0.013). **Conclusions**: A decrease in dynamic valgus angle and an increase in knee flexion angle measured with the DVJ test were noted between the start and the end of the season, suggesting a protective role of the FIFA 11+ training program.

## 1. Introduction

Soccer players are at high risk of knee injuries, particularly those involving anterior cruciate ligament (ACL). ACL injuries can have a detrimental effect on an athlete’s career and overall health, as they involve long recovery times and can lead to subsequent complications such as knee osteoarthritis [1,2]. The annual incidence of ACL tears is 68.6 per 100,000 person-years. Among the mechanisms contributing to these injuries, improper neuromuscular control with excessive dynamic valgus loading of the knee has been identified as one of the main risk factors [3,4,5]. Dynamic knee valgus is a movement pattern of the lower limb that may consist of unfavorable movements of the tibia and femur. Academy players are players under the age of 21 who are registered with and play for a club that manages a youth sector under the rules of professional soccer. Neuromuscular control should be considered, especially in academy soccer players who are going through rapid growth phases and physical development, which may increase their risk of ACL injury [6] with a possible compromise of their professional career.

Different types of training regimens have been examined in pivoting sports and specific neuromuscular strengthening programs have been shown to be superior to standard sports-specific training [7]. Soccer prevention training programs focus on improving strength, coordination, and proprioception and aim to optimize the neuromuscular strategies athletes use in high-risk maneuvers such as cutting, landing, and pivoting, particularly to protect against non-contact ACL injuries [8,9,10]. Among these, the FIFA 11+ program is recommended by the FIFA research center and previous studies support its effectiveness. It is a structured injury prevention program developed specifically for soccer players, aiming to reduce the risk of lower limb injuries, particularly ACL injuries. Unlike classical training, which often focuses on strength, endurance, and technical skills, the FIFA 11+ incorporates dynamic warm-ups, neuromuscular control, proprioception, plyometric exercises, and core stability. These components address biomechanical risk factors associated with ACL injuries, such as poor landing mechanics and valgus knee collapse. Evidence suggests that teams implementing the FIFA 11+ regularly experience a significant reduction in ACL injury rates compared to those following traditional training regimens, highlighting its effectiveness in enhancing athletic performance while prioritizing injury prevention [11,12,13,14]. However, there is limited research on how soccer training programs over an entire season can improve neuromuscular control and reduce the risk of ACL injuries associated with valgus loading in young soccer players.

The drop vertical jump (DVJ) test is an important tool for assessing lower extremity neuromuscular control and ACL injury risk. It measures an athlete’s ability to control the movement of the knee during the landing from a certain height. This ability is directly related to the risk of sustaining an ACL injury [2,15]. In this test, neuromuscular deficits that may contribute to an ACL injury can be observed: (1) increased valgus of the knee on landing; (2) decreased flexion angle on landing; (3) asymmetric landing [16].

There is a lack of data about how neuromuscular programs affect athletes through a season-long period, which necessitates longer follow-ups. Understanding the changes within the sporting season is very important since specific and timely preventive approaches could be introduced for athletes at risk of injury. No study on that matter has been carried out, to our best knowledge. The hypothesis was that at the beginning of the competitive season, athletes would have less neuromuscular control of the lower extremity, with more dynamic knee valgus. As the competitive season comes to an end, neuromuscular control should be higher due to a longer period of training program. This study aimed to analyze the impact of the FIFA 11+ training program on the neuromuscular control and valgus loading of the knee, as measured by the DVJ test, in academy soccer players over a competitive season. Correlations among demographic characteristics and the values of dynamic valgus angle were also investigated to identify soccer players at higher risk of ACL injury.

## 2. Materials and Methods

A prospective study was conducted on 85 male soccer players from the youth sector of two professional sports clubs between August 2022 and June 2023. The study protocol was approved by the local ethics committee (Institutional Review Board approval was obtained from Magna Græcia University of Catanzaro), and the research was conducted in compliance with the declaration of Helsinki. All respondents, or parents in case of age < 18 years old, provided informed consent to participate in the study.

All active soccer players of the youth sector of the professional sports clubs involved in the research were eligible to participate in the study. The level of competition was defined by the Italian football federation [17]. Exclusion criteria were (1) prior ACL reconstruction, or (2) other knee surgery or lower extremity fracture. Thirteen players did not participate in a follow-up. Therefore, 72 of 85 soccer players (85%) were evaluated.

Data gathered included the age of the players, height, weight, limb dominance, and playing position [18].

### 2.1. Drop Vertical Jump Test

This study incorporated the DVJ test to objectively quantify changes in neuromuscular control and valgus loading dynamics as a result of the implemented training regimen. The study was carried out before the match and at the end of the same match at two different times of the season: at the beginning and at the end over a period of 11 months. The first DVJ test was performed before the start of the match. The execution of the DVJ test was demonstrated beforehand to the athletes with a video. To reduce differences during the test allowing a correct evaluation, the subjects were asked to wear shorts and sports shoes. The athletes’ warm-up was executed following FIFA 11+ program [19]; it is a complete dynamic warm-up program created to reduce the incidence of injuries in young soccer players aged 14 or over. It required no equipment other than a ball and was completed in 10–15 min. The exercises focused on core stabilization, eccentric training of thigh muscles, proprioceptive training, dynamic stabilization, and plyometrics with straight leg alignment. The adherence and compliance to the FIFA 11+ program was checked with coaches and soccer players with verbal questions during the study. The second DVJ test was performed immediately after the end of the same match.

The DVJ test consisted of the subject starting on top of a box measuring 31 cm in height, the standard height according to Hewett’s method [16]. Subjects were instructed to drop directly down off the box and immediately perform a maximum vertical jump, raising both arms as if they were jumping for a basketball rebound [20]. Three different videos (Sony FDR-AX33) were simultaneously taken to document the entire execution of the test, in its different phases [21]. Participants were allowed up to 3 practice trials prior to testing. Two trained orthopedic surgeons who were unaware of the patients’ clinical features carried out the assessment.

On the frontal view, two different positions were necessary to evaluate dynamic valgus angles of the knee in the different phases of the test: at the initial contact (IC) with the ground (Figure 1), and at the peak contact with the ground (Figure 2). Negative angular values (i.e., below 0°) indicated knee varus while positive angular values indicated knee valgus on the DVJ test. The positive angular values indicating a dynamic valgus knee were also stratified in three ACL injury risk categories as follow: 0–5° low risk, 5–10° medium risk, >10° high risk. On the right and left lateral views, the flexion angles at the peak contact with the ground were evaluated for the right and left limb, respectively (Figure 3).

### 2.2. Statistical Analysis

All data were collected, measured, and reported at an accuracy of 1 decimal place. The mean, standard deviation, and range are used to present continuous variables, while categorical variables are presented as counts. The distribution of the numeric samples was assessed using the Kolmogorov–Smirnov normality test. Based on this preliminary analysis, parametric tests were adopted. The dynamic valgus angle assessment at the start of the match at the beginning of the season was used to identify the population at risk for ACL injury (≥0° dynamic valgus angle) and to analyze the role of the FIFA 11+ program on this subgroup. Correlations were tested to investigate possible associations among the available data, and Pearson’s coefficient or the phi coefficient was adopted when appropriate. The correlation was considered to be strong (r > 0.5), medium (0.5 < r < 0.3), or small (0.3 < r <0.1). A priori power analysis calculation (G*Power 3.1.9.2 software, Institut fur Experimentelle Psychologie, Heinrich Heine Universitat, Dusseldorf, Germany) showed that a sample size of 36 participants and a medium effect size of 0.25 would provide a power of 0.8. IBM SPSS Statistics software (version 26; IBM Corp., Armonk, NY, USA) and G*Power (version 3.1.9.2; Institut fur Experimentelle Psychologie, Heinrich Heine Universitat, Dusseldorf, Germany) were used to construct the database and perform statistical analysis. A *p*-value of less than 0.05 was considered significant.

## 3. Results

The demographic characteristics of the included athletes are summarized in Table 1. The final sample consisted of 72 patients with a mean age of 17 ± 2.6 years (range, 15–20 years) and a mean body mass index (BMI) of 20.9 ± 2.4 (range, 14.6–28.9).

Table 2 shows the comparison of dynamic valgus angles at IC between the start and the end of the match and between the start and the end of the season. In detail, a statistically significant increase of the varus angle on the right limb was noted between the start and the end of the season in values measured at the beginning of the match (−0.2 ± 3.1 versus −1.2 ± 2.1, *p* = 0.003), suggesting a higher neuromuscular control at the end of the season.

Table 3 shows the comparison of dynamic valgus angles at the peak. In detail, a statistically significant increase of the varus angle on the right limb was noted between the start and the end of the season at the beginning of the match (−4.7 ± 8.9 versus −6.9 ± 6, *p* = 0.003) and between the start and the end of the match in values measured at the beginning of the season (−4.7 ± 8.9 versus −7.7 ± 9, *p* < 0.001). A statistically significant decrease of the varus angle on the left limb was noted between the start and the end of the season in values measured at the beginning of the match (−9.9 ± 6.9 versus −7.6 ± 6.7, *p* = 0.003) and between the start and the end of the match in values measured at the beginning of the season (−9.9 ± 6.9 versus −8 ± 8, *p* = 0.012).

Table 4 shows the comparison of flexion knee angles at the peak. In detail, a statistically significant increase of the flexion angle on both limbs was noted between the start and the end of the season in values measured at the start of the match (left limb 76.8 ± 32.8 versus 98.6 ± 17.2, *p* <0.001; right limb 76.4 ± 32.8 versus 96.1 ± 16.1, *p* <0.001) and between the start and the end of the season in values measured at the end of the match (left limb 92.8 ± 19.1 versus 98.5 ± 16.3, *p* = 0.002; right limb 92.6 ± 19.2 versus 96.7± 14.5, *p* = 0.013).

Considering the subjects at risk for ACL injury according to the dynamic valgus angle assessed at the start of the match at the beginning of the season, Table 5 shows the comparison of dynamic valgus angles at IC between the start and the end of the match and between the start and the end of the season. In detail, a statistically significant increase of the varus angle on the right limb was noted between the start and the end of the season in values measured at the beginning of the match (2.7 ± 1 versus 0.9 ± 2, *p* = 0.031).

Table 6 shows the comparison of dynamic valgus angles at the peak; no statistically significant differences were reported.

Table 7 shows the comparison of flexion knee angles at the peak. In detail, a statistically significant increase of the flexion angle on both limbs was noted between the start and the end of the season in values measured at the start of the match (left limb 71.4 ± 5.6 versus 85.2 ± 12.6, *p* < 0.001; right limb 70.6 ± 5 versus 83.2 ± 12.2, *p* < 0.001) and between the start and the end of the season in values measured at the end of the match (left limb 74.1 ± 12.3 versus 87.4 ± 12.7, *p* < 0.001; right limb 74.8 ± 12 versus 88.9 ± 12.7, *p* < 0.001), suggesting, overall, a higher neuromuscular control at the end of the season and also among the subjects at risk for ACL injury. Cohen’s kappa coefficients for intraobserver and interobserver reliability of results at DVJ test were 0.88 and 0.86, respectively.

Table 8 shows the results of correlations among age, height, weight, playing position, and the values of dynamic valgus angles at IC, dynamic valgus angles at the peak, and flexion knee angles at the peak. Higher age, height, and weight correlated with a greater dynamic knee valgus and flexion angles.

## 4. Discussion

The most important finding of the present study was that a decrease in dynamic valgus angle and an increase in knee flexion angle measured with the DVJ test was noted between the start and the end of the season in academy soccer players trained with the FIFA 11+ program. The change in dynamic valgus and knee flexion angles was also present when only subjects at risk for ACL injury were considered, suggesting a protective role of the training program regardless of baseline condition. Higher age, height, and weight correlated with greater dynamic valgus and flexion angles of the knee.

Playing soccer requires various abilities and skills, including agility, speed, endurance, and a technical and tactical understanding of the game. All these aspects are taught and improved during training sessions [22]. However, playing soccer also entails a substantial risk of injury, and an optimal training program should include exercises to reduce this risk.

In the current study, a decrease in dynamic valgus angle and an increase in knee flexion angle assessed with the DVJ test was noted between the start and the end of the season. This observation was made both at initial contact and at peak contact with the ground, confirming the hypothesis that neuromuscular control is higher as the season draws to a close. As dynamic valgus loading during the DVJ is a known risk factor for ACL injury, at the end of the season the risk of ACL injury is lower [3]. In addition, the knee flexion angle tends to be higher during the DVJ test performed at the end of the season. Interestingly, a greater knee flexion angle is a protective factor against a possible ACL injury [20]. Although fatigue may play a role in increasing ACL injury risk, most ACL injuries occur early in the competitive season in soccer, basketball, and lacrosse [23].

The decrease in values of dynamic valgus over the competitive season was also present when only subjects at risk for ACL injury were considered. These findings confirm the protective role of season-long prevention training programs with a higher neuromuscular control achieved at the end of the soccer season regardless of the baseline condition.

Overall, the differences observed between the start and the end of the season in terms of dynamic valgus angle were 2–3° in all statistically significant comparisons. It should be noted that this difference, albeit minimal, allowed reaching an angular value closer to that of safety or low risk for ACL injury considering the three risk categories according to the DVJ test. The gain in knee flexion was also clinically significant. In fact, a 20° increase was observed between the start and end of the season, which allowed athletes to land with a knee flexed at 90° at the end of the season, further protecting against the risk of ACL injury.

In recent years, interest in ACL injury prevention programs has increased. Different types of training programs have been studied for different sports and specific neuromuscular strengthening programs have been shown to be superior to normal sport-specific training [7]. The FIFA 11+, a program developed by the FIFA research center in 2006, was conducted before each training session and match in the sports clubs investigated in the current study. It is a complete warm-up program designed to reduce the incidence of sports injuries in soccer players and includes running exercises as well as core and leg strength exercises. The exercises included in the FIFA 11+ program lead to a strengthening of the muscles, and in addition, static, dynamic and reactive neuromuscular control, balance, agility, coordination, and jump technique are improved. The efficacy of FIFA 11+ was first proven in young female players as was the Prevent and Enhance Performance program, a non-contact ACL prevention program. A randomized controlled trial investigated how the program might impact the rate of ACL injury in male soccer players, reporting a 4.25-fold reduction in the likelihood of incurring ACL injury. Recently, a study on youth recreational soccer players also reported that FIFA 11+ was adequate to detect positive changes in landing biomechanical markers of ACL injury. Previous studies also support the efficacy of this program in reducing other injuries (i.e., groin and thigh strains as well as ankle sprains) [11,12,13,14]. A meta-analysis by Thorborg et al. confirmed that FIFA 11+ has a significant injury-reducing effect of 39% [24].

In the current study, correlation analysis among demographic characteristics and the values of dynamic valgus angles was performed to identify soccer players at higher risk of ACL injury. We reported that higher age, height, and weight correlated with greater dynamic knee valgus and higher flexion angles. Greater knee valgus during the jump test is the predisposing risk factor for ACL injury, whereas greater knee flexion decreases the possibility of ACL injury, suggesting a compensatory mechanism [3,20,25]. Indeed, there should be consideration of the importance of avoiding an extended knee landing position, which is a known risk factor for ACL injury, especially with a concomitant increased tibial slope. Regarding the role of age, a recent systematic review did not find age to be a risk factor for ACL injury in soccer [26,27,28], as is the case for other sports-related injuries [28,29]. However, a significant age dependency in the effectiveness of neuromuscular training regimens to reduce ACL injuries has been reported [30]. There is no clear evidence in the literature that body height affects the ACL injury incidence [31]. On the other hand, a study showed that underage male players who reached the pubertal growth spur later had worse results in the DVJ [29]; in addition, a higher BMI was reported to be a risk factor for a possible ACL injury [32]. Therefore, soccer players with these anthropometric characteristics should be subjected to more careful monitoring by coaches and trainers during the season.

Our study has several limitations that should be considered. First, the number of soccer players might have been small to detect meaningful differences, but a power of 0.8 was satisfied by the actual sample size as calculated. Another limitation was that all soccer players in this study were males, so further studies to assess the valgus loading of the knee in females would be indicated to explore possible gender differences. The playing time of each player during the matches was not recorded and this could have influenced the results of the DVJ test. In this context, the lack of a fatigue marker could also have altered the angular values detected. However, we have included all the active soccer players in the squad to obtain the broadest and most naturalistic finding, also considering the collection of data on a homogenous sample and over a long period. It should be also considered that DVJ test learning effects might have occurred; soccer players may have achieved better results when they repeated the test due to feed forward mechanisms with possible adjusting behavior as pre-programmed neural activation in preparing muscle activity to maximize contact forces and store elastic energy in the musculotendinous structure during the next eccentric phase [33]. There are different modifications of the jump tests to measure dynamic valgus of the knee, like single leg jump, tuck jump. Still, the DVJ test is considered the golden standard in dynamic knee valgus testing [15,25]. Differences in dynamic valgus angles were often small and may not be clinically relevant although statistically significant. The lack of administration of a functional outcome scale and specific test of neuromuscular control represent additional weaknesses. Moreover, the role of other biomechanical factors such as the tibial slope has not been investigated. However, the prospective nature of the data collection, the use of the FIFA 11+ program, and the use of validated, accurate, and standardized dynamic knee valgus assessments with high values for intraobserver and interobserver reliability represent considerable strengths of the present study. Considering the ethics committee view and the opinion of this study group, and knowing the benefits of FIFA 11+ program, it would be deemed unethical to have a group of young athletes being left aside as a control group, putting them in danger of a possible debilitating injury. However, a control group would have allowed for a more direct comparison of the effects of the training program and would have strengthened the conclusions drawing a causal effect. In future research, the FIFA 11+ program could be compared to other programs of neuromuscular strengthening.

The FIFA 11+ program has been shown to be effective in decreasing dynamic knee valgus, which is a known risk factor for ACL tears. The protective role of this training program should be considered by coaches and health professionals involved in the management of young soccer players. The implementation of the DVJ test between the medical and physical examinations at the beginning and end of the season allowed identification of subjects with dynamic knee valgus values that place them at risk for ACL injury; even in these soccer players, the protective role of the FIFA 11+ program was detected, favoring its use. Finally, particular attention should be paid to taller and heavier soccer players who appear to have a greater risk of ACL injury.

## 5. Conclusions

A decrease in dynamic valgus angle and an increase in knee flexion angle measured with the DVJ test were noted between the start and the end of the season in academy soccer players trained with the FIFA 11+ program. The change in dynamic valgus and knee flexion angles was also present when only subjects at risk for ACL injury were considered, suggesting a protective role of the training program regardless of baseline condition.

## Figures and Tables

**Figure 1 healthcare-13-00073-f001:**
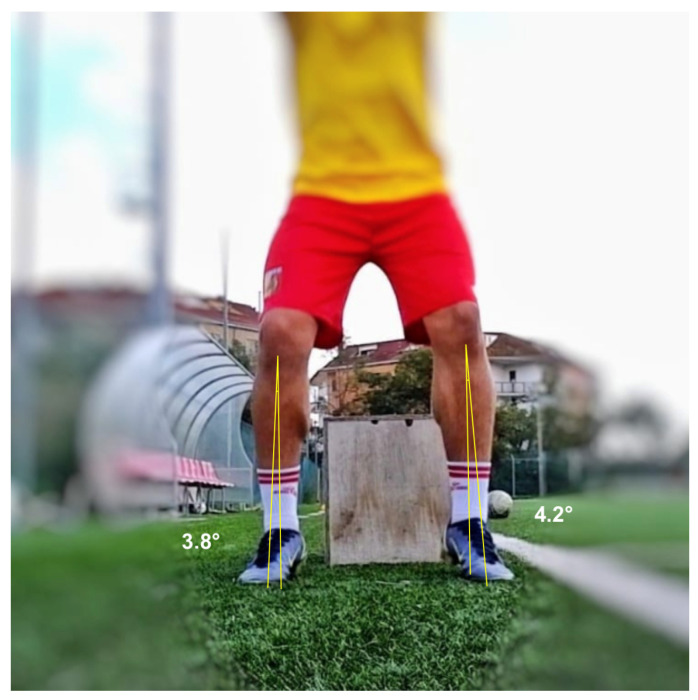
Dynamic valgus angles measured at the initial contact with the ground.

**Figure 2 healthcare-13-00073-f002:**
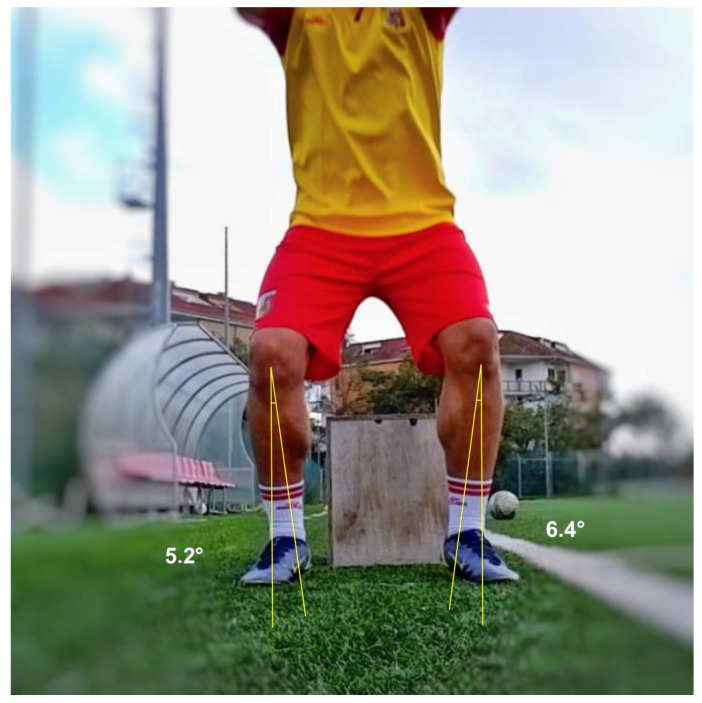
Dynamic valgus angles measured at the peak contact with the ground.

**Figure 3 healthcare-13-00073-f003:**
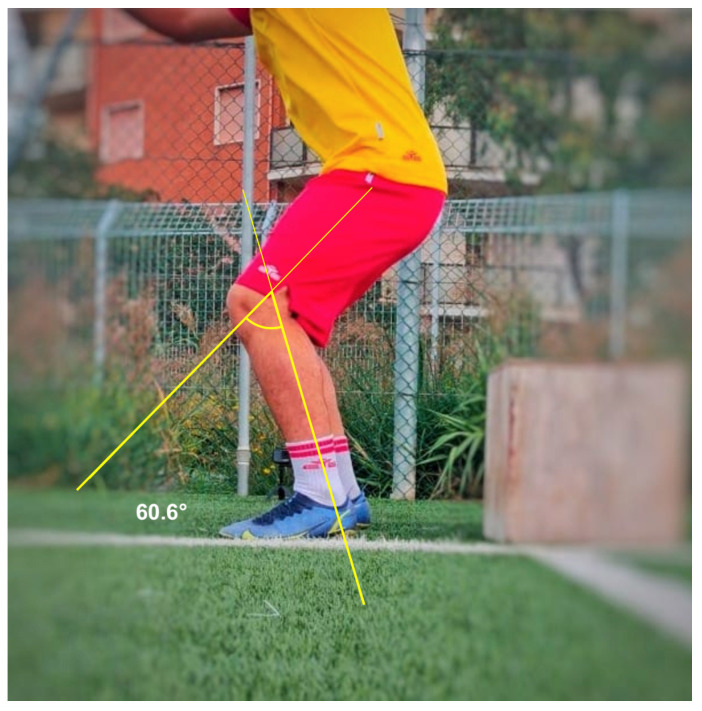
Flexion knee angle measured at the peak contact with the ground.

**Table 1 healthcare-13-00073-t001:** Baseline characteristics of included patients.

Patients (No. = 72)	Mean ± SD (Range) or No. (%)
Gender	
Male	72 (100%)
Age (year)	17 ± 2.6 (15–20)
Height (m)	1.7 ± 0.1 (1.5–2)
Weight (kg)	62.7 ± 11.1 (36.4–84.7)
BMI (kg/m^2^)	20.9 ± 2.4 (14.6–28.9)
Right dominant limb	56 (77.8%)
Left dominant limb	16 (22.2%)
Position	
Goalkeeper	7 (9.7%)
Defender	26 (36.1%)
Midfielder	24 (33.4%)
Forward	15 (20.8%)

BMI means body mass index; SD, standard deviation; No., number of cases.

**Table 2 healthcare-13-00073-t002:** Comparison of dynamic valgus angles at initial contact between the start and the end of the match and between the start and the end of the season.

Start of the Season	Limb	Mean	SD		End of the Season	Limb	Mean	SD		*p*-Value Start vs. End of the Season	95% CI	SED
start of the match	left	−1.7	2.5		start of the match	left	−1.4	2.2		0.482	−0.96 to 0.46	0.354
	right	−0.2	3.1			right	−1.2	2.1		**0.003**	0.33 to 1.54	0.303
end of the match	left	−1.8	2.6		end of the match	left	−1.8	2.1		0.963	−0.59 to 0.62	0.302
	right	−0.7	2.4			right	−0.7	2		0.961	−0.58 to 0.56	0.286
		*p*-value start vs. end of the match	95% CI	SED			*p*-value start vs. end of the match	95% CI	SED			
	left	0.719	−0.63 to 0.91	0.385		left	0.081	−0.05 to 0.86	0.228			
	right	0.198	−0.26 to 1.23	0.374		right	0.101	−1.01 to 0.09	0.276			

SD, standard deviation; CI, confidence interval; SED, standard error of difference. *p*-Value < 0.05 are in bold.

**Table 3 healthcare-13-00073-t003:** Comparison of dynamic valgus angles at the peak between the start and the end of the match and between the start and the end of the season.

Start of the Season	Limb	Mean	SD		End of the Season	Limb	Mean	SD		*p*-Value Start vs. End of the Season	95% CI	SED
start of the match	left	−9.93	6.85		start of the match	left	−7.64	6.74		**0.003**	−3.78 to −0.80	0.747
	right	−4.67	8.85			right	−6.92	5.96		**0.003**	0.77 to 3.73	0.741
end of the match	left	−8.01	7.95		end of the match	left	−6.75	7.16		0.087	−2.72 to 0.19	0.728
	right	−7.72	8.95			right	−6.50	6.92		0.235	−3.26 to 0.81	1.020
		*p*-value start vs. end of the match	95% CI	SED			*p*-value start vs. end of the match	95% CI	SED			
	left	**0.012**	−3.41 to −0.42	0.748		left	0.166	−2.16 to 0.38	0.635			
	right	**<0.001**	1.59 to 4.52	0.734		right	0.567	−1.86 to 1.03	0.725			

SD, standard deviation; CI, confidence interval; SED, standard error of difference. *p*-Value < 0.05 are in bold.

**Table 4 healthcare-13-00073-t004:** Comparison of flexion knee angles at the peak between the start and the end of the match and between the start and the end of the season.

Start of the Season	Limb	Mean	SD		End of the Season	Limb	Mean	SD		*p*-Value Start vs. End of the Season	95% CI	SED
start of the match	left	76.97	32.78		start of the match	left	98.63	17.24		**<0.001**	−30.47 to −12.84	4.422
	right	76.36	32.79			right	96.13	16.13		**<0.001**	−28.34 to −11.18	4.303
end of the match	left	92.75	19.10		end of the match	left	98.47	16.28		**0.002**	−9.27 to −2.17	1.780
	right	92.57	19.24			right	96.71	14.45		**0.013**	−7.39 to −0.89	1.628
		*p*-value start vs. end of the match	95% CI	SED			*p*-value start vs. end of the match	95% CI	SED			
	left	**0.001**	−25.05 to −6.50	4.653		left	0.898	−2.24 to 2.54	1.198			
	right	**<0.001**	−25.25 to −7.17	4.534		right	0.620	−2.92 to 1.75	1.171			

SD, standard deviation; CI, confidence interval; SED, standard error of difference. *p*-Value < 0.05 are in bold.

**Table 5 healthcare-13-00073-t005:** Comparison of dynamic valgus angles at initial contact between the start and the end of the match and between the start and the end of the season among the subjects at risk for ACL injury.

Start of the Season	Limb	Mean	SD		End of the Season	Limb	Mean	SD		*p*-Value Start vs. End of the Season	95% CI	SED
start of the match	left	1.22	2.82		start of the match	left	0,89	1.96		0.774	−2.09 to 2.76	1.145
	right	2.67	1			right	0.89	2.03		**0.031**	0.18 to 3.38	0.754
end of the match	left	0	1.80		end of the match	left	−0.67	2.24		0.496	−1.36 to 2.70	0.957
	right	2	1.94			right	0.78	2.05		0.212	−0.77 to 3.21	0.940
		*p*-value start vs. end of the match	95% CI	SED			*p*-value start vs. end of the match	95% CI	SED			
	left	0.289	−1.14 to 3.59	1.115		left	0.137	−0.55 to 3.66	0.992			
	right	0.372	−0.87 to 2.21	0.726		right	0.909	−1.93 to 2.15	0.961			

SD, standard deviation; CI, confidence interval; SED, standard error of difference. *p*-Value < 0.05 are in bold.

**Table 6 healthcare-13-00073-t006:** Comparison of dynamic valgus angles at the peak between the start and the end of the match and between the start and the end of the season among the subjects at risk for ACL injury.

Start of the Season	Limb	Mean	SD		End of the Season	Limb	Mean	SD		*p*-Value Start vs. End of the Season	95% CI	SED
start of the match	left	6.33	5.86		start of the match	left	1	5		0.297	−7.01 to 17.68	4.447
	right	4	1			right	1	4.58		0.330	−4.52 to 10.52	2.708
end of the match	left	1.67	6.66		end of the match	left	4	1		0.580	−13.13 to 8.46	3.887
	right	0.33	3.79			right	−1	5.29		0.740	−9.10 to 11.76	3.756
		*p*-value start vs. end of the match	95% CI	SED			*p*-value start vs. end of the match	95% CI	SED			
	left	0.413	−9.55 to 18.88	5.121		left	0.366	−11.17 to 5.17	2.944			
	right	0.180	−2.61 to 9.94	2.261		right	0.646	−9.22 to 13.22	4.041			

SD, standard deviation; CI, confidence interval; SED, standard error of difference.

**Table 7 healthcare-13-00073-t007:** Comparison of flexion knee angles at the peak between the start and the end of the match and between the start and the end of the season among the subjects at risk for ACL injury.

Start of the Season	Limb	Mean	SD		End of the Season	Limb	Mean	SD		*p*-Value Start vs. End of the Season	95% CI	SED
start of the match	left	71.38	5.55		start of the match	left	85.19	12.6		**<0.001**	−19.39 to −8.23	2.677
	right	70.57	5.03			right	83.19	12.21		**<0.001**	−18.24 to −7.00	2.693
end of the match	left	74.14	12.26		end of the match	left	87.38	12.73		**<0.001**	−20.51 to −5.96	3.488
	right	74.81	12.04			right	88.86	12.67		**<0.001**	−17.61 to −6.48	2.668
		*p*-value start vs. end of the match	95% CI	SED			*p*-value start vs. end of the match	95% CI	SED			
	left	0.234	−7.46 to 1.93	2.251		left	0.402	−7.53 to 3.14	2.56			
	right	0.097	−9.32 to 0.84	2.437		right	0.139	−8.63 to 1.30	2.380			

SD, standard deviation; CI, confidence interval; SED, standard error of difference. *p*-Value < 0.05 are in bold.

**Table 8 healthcare-13-00073-t008:** Correlation analysis among demographic characteristics and dynamic valgus and flexion knee angles.

	Dynamix Valgus Angle at Initial Contact		Dynamix Valgus Angle at the Peak		Flexion Knee Angle at the Peak	
	Left	Right	Left	Right	Left	Right
Age	0.110	0.030	0.016	0.136	0.946	0.939
	**(*p* = 0.005)**	(*p* = 0.146)	(*p* = 0.288)	**(*p* < 0.001)**	(*p* = 0.540)	**(*p* < 0.001)**
Height	0.615	0.107	0.508	0.122	0.922	0.917
	**(*p* < 0.001)**	(***p* = 0.005**)	(***p* = 0.001**)	**(*p* < 0.001)**	**(*p* < 0.001)**	**(*p* < 0.001)**
Weight	0.610	0.003	0.099	0.033	0.188	0.223
	**(*p* < 0.001)**	(*p* = 0.799)	**(*p* = 0.001)**	(***p* = 0.029**)	**(*p* < 0.001)**	**(*p* < 0.001)**
Position	0.462	0.166	0.459	0.124	0.764	0.723
	**(*p* < 0.001)**	(*p* = 0.939)	**(*p* < 0.001)**	**(*p* < 0.001)**	**(*p* < 0.001)**	**(*p* < 0.001)**
Flexion knee angle at the peak	0.105	0.025	0.552	0.202		
	(*p* = 0.176)	(*p* = 0.216)	**(*p* < 0.001)**	**(*p* < 0.001)**		

*p*-Value < 0.05 are in bold.

## Data Availability

Data are contained within the article.

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
