# Peer review of "The Protective Role of the FIFA 11+ Training Program on the Valgus Loading of the Knee in Academy Soccer Players Across a Season"

_healthcare, 2025, doi:10.3390/healthcare13010073_

Round 1

Reviewer 1 Report

Comments and Suggestions for Authors

Dear Authors,

I have carefully reviewed the manuscript entitled "The protective role of the FIFA 11 + training program on the valgus loading of the knee in academy soccer players across a season" (Manuscript Number: healthcare-3341203).

Overall, I found the manuscript to be well-written and to present a novel contribution to the field of prevention of ACL injury. However, I would like to suggest a few revisions to enhance the clarity and impact of the work.

Specific Comments:

Introduction: While the introduction provides a good overview of the research topic, I suggest revising its structure for better clarity. The current two-paragraph format, with a particularly lengthy second paragraph, could be improved by breaking down the information into more concise paragraphs. This would help the reader follow the logical progression of ideas more easily.

Experimental Design: The choice of the DVJ task as a control condition raises some questions. While the DVJ task is a well-established measure, a more specific soccer-specific task might have provided a more direct comparison to the experimental condition and allowed for more robust generalizations to real-world football performance. Please elaborate on the rationale for choosing the DVJ task and discuss the potential limitations of this choice.

Tables: The presentation of the results in Tables 2-7 could be improved. I recommend combining these tables into a single table to enhance readability and facilitate comparisons. Additionally, consider revising the table format to highlight the most important findings.

Control Group: The absence of a control group that performed standard soccer warm-up routines is a notable limitation of the study. A control group would have allowed for a more direct comparison of the effects of the experimental intervention and would have strengthened the conclusions. Please discuss the reasons for not including a control group and suggest potential avenues for addressing this limitation in future research.

I believe that with the suggested revisions, this manuscript has the potential to make a significant contribution to the field of ACL injury prevention. I recommend addressing the comments raised in this review.

Thank you for the opportunity to review this manuscript.

Sincerely,

Reviewer 2 Report

Comments and Suggestions for Authors

General Comments:

This manuscript addresses an important and timely topic by investigating the effect of the FIFA 11+ training program on knee valgus loading in academy soccer players. The research question is highly relevant, and the methodology is solid in many respects. However, there are areas where the manuscript could be significantly improved to enhance clarity, rigor, and overall impact. Below, I provide a detailed critique with suggestions for improvement.

1.     Abstract: Minor typos and formatting error: Line 17: "Backgroung" should be corrected to "Background."

2.     Introduction

  • The introduction lays a good foundation but could benefit from being more concise. For example, the discussion on dynamic knee valgus (Lines 41–50) includes some redundant details that could be streamlined to improve readability.
  • The novelty of the study—focusing on seasonal changes in knee dynamics—is interesting but underemphasized. Highlighting this gap more explicitly would strengthen the argument for the study’s importance.
  • While the hypothesis is stated, it should be more explicitly linked to the identified research gaps to create a clearer rationale for the study.

3.     Methods

  • The description of the Drop Vertical Jump (DVJ) test (Lines 92–121) is overly detailed, overshadowing other critical aspects of the methodology. Condensing this section would free up space to provide more details on participant recruitment and the implementation of the FIFA 11+ program.
  • The choice of the 31 cm box height for the DVJ test is not justified. Since box height can influence results, it would be helpful to explain why this specific height was chosen.
  • Adherence to the FIFA 11+ program across the season is not addressed. This is an important limitation—was compliance monitored? Without this information, the study's validity could be questioned.

4.     Results

  • The results are thorough but presented in a way that feels dense and, at times, repetitive. For instance, comparisons across Tables 2–7 could be consolidated to improve readability.
  • While statistical significance is reported, the manuscript does not delve deeply into the clinical relevance of the findings. For example, what is the practical implication of the observed "small differences" discussed in the manuscript? Addressing this would make the findings more meaningful.
  • Table 8 presents correlation analyses, but these are not sufficiently tied back to the study’s primary hypothesis. A clearer interpretation of these associations is needed to strengthen the narrative.
  • Tables lack meaningful interpretation in the text. Instead of reporting all p-values, focus on those critical to the hypothesis.

5.     Discussion

  • The discussion largely restates findings without offering deeper insights. For example, why might knee flexion angles increase over the season? Are these results consistent with findings from other studies? Engaging more critically with the results would enrich the discussion.
  • Fatigue is mentioned as a potential confounder (Line 248), but the methodology does not address it. The lack of fatigue markers is a notable gap that undermines the argument. Including a discussion of how this limitation could affect the results would improve transparency.
  • While limitations are acknowledged, they could be expanded further:
    • The lack of a control group limits the ability to draw causal conclusions.
    • Learning effects from repeated DVJ tests are mentioned but not sufficiently discussed.
    • The study focuses exclusively on male athletes, which misses an opportunity to explore gender differences in neuromuscular control.

6.    References

  • Many of the cited studies (e.g., Lines 333–336) are older. Adding more recent literature would make the manuscript more relevant and up-to-date.
Comments on the Quality of English Language

While the manuscript's english quality does not restrict comprehension, a professional language edit would refine its presentation. This would ensure grammatical precision, consistency in tone and voice, and improved clarity, ultimately enhancing the manuscript's readability and academic impact.

Reviewer 3 Report

Comments and Suggestions for Authors

This study is a confirmatory one carried out on a small number of athletes.

The research only analyzes measurements in different stages of sports training.

It was interesting if, after the first series of analyzed measurements, there would have been intervention in the training in order to correct the identified deficiencies.

Reviewer 4 Report

Comments and Suggestions for Authors

In the current study, authors aimed to analyze the impact of the FIFA 11+ training program on the valgus loading of knees in academy soccer players over a competitive season. It is well known that rapid growth phases and physical development may increase their risk of ACL injury, and this situation may compromise their professional careers.

Therefore, studies focusing on the healthy physical  and professional career development of young players are very valuable. In this respect, I congratulate the authors for their study.

The study generally has an academic design and language. The suggestions that I think would contribute to the study are stated below.

In the introduction, the frequency of ACL injuries due to classical training is mentioned by providing data.

- Mention the general content of the FIFA 11+ training program, its difference from classical training, and its relationship with ACL injuries.

- Indicate the brand and origin of the camera used.

- Provide detailed information about the FIFA 11+ program applied in the training in the methodology.

- Reorganize your tables in APA format.

- In the discussion section, you mentioned the FIFA 11+ training program. Add this section to the introduction and methodology section, and discuss the program's effect on ACL injuries in this section based on references.

- Since the classical training program and the FIFA 11+ program were not compared in your study, in the discussion section, provide data from the literature on ACL injuries in young football players and discuss the application of the FIFA 11+ program and its relationship with your data.

Round 2

Reviewer 1 Report

Comments and Suggestions for Authors

Dear Authors,

I have carefully reviewed the revised version of manuscript entitled "The protective role of the FIFA 11 + training program on the valgus loading of the knee in academy soccer players across a season" (Manuscript Number: healthcare-3341203).

Overall, I found the manuscript have improved. However, I would like to combining these tables into a single table to enhance readability and facilitate comparisons.

Author Response

Dear Authors,

I have carefully reviewed the revised version of manuscript entitled "The protective role of the FIFA 11 + training program on the valgus loading of the knee in academy soccer players across a season" (Manuscript Number: healthcare-3341203).

Overall, I found the manuscript have improved. However, I would like to combining these tables into a single table to enhance readability and facilitate comparisons.

A: Thank you for your comment. Thanks to your valuable suggestions the manuscript has improved significantly. We thank you for the time and attention you have given us. We have also previously revised the format of the tables as recommended. In terms of creating a single table, we think it is not so easy to create without losing the data reported. By the way, we reserve the right to ask for kind support from the Editorial Staff.

Reviewer 2 Report

Comments and Suggestions for Authors

The manuscript presents a compelling analysis of the FIFA 11+ program's impact on neuromuscular control and knee valgus loading among academy soccer players. The methodology and statistical rigor are commendable, but improved clarity, additional references, and expanded explanations could benefit some areas.

Comments for Authors:

Ensure the introduction clearly highlights the study's unique contributions to the existing research on ACL injury prevention.

Address the absence of a control group in the discussion section to preempt potential criticisms.

Provide additional context for the small but statistically significant changes in valgus and flexion angles.

Consider revising the text for clarity and conciseness to improve overall readability.

Comments on the Quality of English Language

Some sentences are overly complex or lack clarity. Simplifying the language would enhance readability.

Author Response

Ensure the introduction clearly highlights the study's unique contributions to the existing research on ACL injury prevention.

A: Thanks to your valuable suggestions the manuscript has improved significantly. We thank you for the time and attention you have given us. We highlighted the study's unique contributions in the introduction as suggested

Line 96-98: Understanding the changes within the sporting season is very important since specific and timely preventive approaches could be introduced for athletes at risk of injury.

Address the absence of a control group in the discussion section to preempt potential criticisms.

A: Thank you for your comment. We addressed the absence of a control group in the discussion section as suggested.

Line: 430-434: However, a control group would have allowed for a more direct comparison of the effects of the training program and would have strengthened the conclusions drawing a causal effect. In future research, FIFA11+ program could be compared to other programs of neuromuscular strengthening.

Provide additional context for the small but statistically significant changes in valgus and flexion angles.

A: Thank you for your comment. We provided additional information as suggested.

Line 350-357: Overall, the differences observed between the start and the end of the season in terms of dynamic valgus angle were 2-3° in all statistically significant comparisons. It should be noted that this difference, albeit minimal, allowed to reach an angular value closer to that of safety or low risk for ACL injury considering the three risk categories according to the DVJ test. The gain in knee flexion was also clinically significant. In fact, a 20° increase has been observed between the start and end of the season which allowed athletes to land with a knee flexed at 90° at the end of the season further protecting against the risk of ACL injury.

Consider revising the text for clarity and conciseness to improve overall readability.

A: Thank you for your comment. We revised the text as recommended.